# The Swab, the Drip, or the Meat? Comparison of Microbiological Sampling Methods in Vacuum-Packed Raw Beef

**DOI:** 10.3390/microorganisms13010159

**Published:** 2025-01-14

**Authors:** Aracely Martínez-Moreno, America Chávez-Martínez, Janet E. Corry, Christopher R. Helps, Raúl. A. Reyes-Villagrana, Juan. M. Tirado Gallegos, Eduardo Santellano-Estrada, Ana L. Rentería-Monterrubio

**Affiliations:** 1UACH-CA03 Tecnología de Alimentos de Origen Animal, Facultad de Zootecnia y Ecología, Universidad Autónoma de Chihuahua, Periférico Fco. R. Almada, Chihuahua 33820, Mexicoamchavez@uach.mx (A.C.-M.); jtirado@uach.mx (J.M.T.G.); esantellano@uach.mx (E.S.-E.); 2Department of Clinical Veterinary Science, University of Bristol, North Somerset, Langford BS40 5DU, UK; 3Researcher CONAHCYT, Av. Insurgentes Sur, No 1582, Col, Crédito Constructor, Mexico City 03940, Mexico

**Keywords:** meat microbiology, microbial sampling, swabbing, excision, drip, vacuum-packed meat

## Abstract

Historically, there has been a concern for the detection and enumeration of microorganisms in foods, and numerous methods have been developed to determine their microbiological conditions. The present study aimed to compare the numbers of microbes recovered with three sampling methods: drip, excision, and swabbing in vacuum-packed beef. The sampling methods were evaluated in terms of the viable numbers of *Enterobacteriaceae*, lactic acid bacteria (LAB), *Brochrothrix thermosphacta*, *Salmonella* spp., and yeasts and moulds (Y&M). The numbers of *B. thermosphacta*, *Salmonella* spp., *Enterobacteriaceae*, LAB, and M&Y recovered with the drip method were significantly higher (*p* < 0.05) than those from the other two methods. Regarding excision and swabbing, the recovery of *B. thermosphacta* and *Enterobacteriaceae* was higher (*p* < 0.05) with the excision method than swabbing, while there were no statistical differences (*p* > 0.05) between both methods for *Salmonella* spp., LAB, and Y&M. In conclusion, the drip method can recover up to two logarithms more than the other techniques in vacuum-packed meat; hence, it should be considered when designing and implementing sampling systems for the meat industry.

## 1. Introduction

Foods of animal origin (FOAO) and their by-products have a natural microbial population, since animals are reservoirs for a wide range of microorganisms [1]. Meanwhile, FOAO are essential for the supply of protein in the daily diet [2]. However, if these are contaminated with pathogens, their consumption risks the health of the consumer [3,4]. Also, the economic losses associated with spoilage microorganisms are a problem for consumers, the meat industry, and governments [3,4]. Historically, there has been a concern with the detection and enumeration of microorganisms in foods, particularly meat, and therefore, numerous methods have been developed to determine their microbiological status [5]. National and international regulations detail the methodologies for verifying the presence or absence of pathogens, indicators, and spoilage microorganisms. For example, the 17,604 standard of the International Organization for Standardization [6] for meat and meat products details the specifications for carcass sampling and microbiological analyses. Sampling methods are classified as invasive (excision) and non-invasive (rinse, sponge, and swab), and the choice of method will depend on the objective of the analysis, sensitivity, and practical considerations [6,7,8]. Metrology, the science of measurements, establishes the standards and procedures to conduct more rigorous experiments to obtain the most accurate measurements, as established by the International Bureau of Weights and Measures (Bureau International des Poids et Mesures, BIPM). These include the International System of Units, which is essential for the unification of measurement standards and conversion factors. In this context, microbiology reviews and traces the growth of the microorganisms, focusing on the density of the population in terms of colony-forming units, either per unit of area or volume, i.e., CFU/cm^2^ or CFU/mL. However, it is necessary to have a mathematical model that facilitates the conversion of units, where the study parameter is density, in general. Therefore, this study aimed to compare the number of microorganisms recovered from vacuum-packed beef following the swabbing, excision, and drip methods.

## 2. Materials and Methods

### 2.1. Meat Samples

Approximately 20 kg of fresh beef (*Semimembranosus* muscle, from meat producers Beef International SA de CV (Chihuahua, Mexico)) were sliced into 1 cm thick fillets (~450 g each). Then, each fillet was vacuum-packed and aged for 72 h at 4 ± 2 °C, so that the meat would release the exudate.

### 2.2. Microbiological Sampling

After the ageing period, all fillets were sampled using the three protocols: swabbing (SW), excision (EX), and drip (DP), always taking the DP sample first. Before opening the packages, these were disinfected with 70% ethyl alcohol.

For the DP samples, the packs were held vertically so that the exudate was collected in one corner. Subsequently, using a sterile scalpel blade, the bags were opened, and the exudate was removed with a sterile Pasteur pipette and placed in a sterile vial. The collection of SW and EX surface samples were taken with a sterile stainless-steel template (25 cm^2^), which was randomly placed in different areas of the sample, avoiding fat and connective tissue.

In the SW samples, a 25 cm^2^ area was rubbed with a sterile swab with horizontal and vertical moves until it was completely covered. Subsequently, the swab was placed in 10 mL of sterile diluent (maximum recovery diluent, MRD, CM0733, Oxoid, Basingstoke, UK) and homogenized with a vortex (Vortex Genie-2, Scientific Industries, Bohemia, NY, USA) for 60 s.

For the EX method, the template was pressed firmly onto the meat, and the protruding area was cut approximately 2 mm deep. The template was then removed, and a 1–2 mm thick layer of meat was taken from the delimited area. The meat sample was placed in a Stomacher^®^ bag (BA6040, Seward, Sussex, UK) with 9 mL of sterile diluent and homogenized for 1 min in a Stomacher^®^ 80 Biomaster (Seward, AK, USA). The homogenized samples were allowed to stand for 1 min for large particles to settle.

### 2.3. Microbiological Enumeration

The samples (SW, EX, DP) were serially diluted (1:10–1:1,000,000) in sterile diluent, and 10 μL of each dilution were streaked onto specific culture agars in duplicates. Lactic acid bacteria (LAB) were plated onto de Man, Rogosa Sharpe agar (MRS, CM0361, Oxoid^®,^ Basingstoke, UK), moulds and yeasts (M&Y) on Rose Bengal Chloramphenicol agar (RBC, CM0549, chloramphenicol supplement, SR0078, Oxoid^®^, Basingstoke, UK), *Salmonella* and *Shigella* on Salmonella Shigella agar (SS, CM0099, Oxoid^®^, Basingstoke, UK), *Enterobacteriaceae* on MacConkey agar (MCK, BO0550 Oxoid^®^, Basingstoke, UK), and *Brochothrix thermosphacta* on Streptomycin Thallium Acetate Actidione agar (STAA, CM0881, Oxoid©, Basingstoke, UK). The media were incubated aerobically at 35 °C except for *B. thermosphacta*, which was incubated at 25 °C, and the LAB, which were incubated under anaerobic conditions. Colony-forming units (CFUs) were counted on plates containing between 10 and 200 colonies, and the results were converted to logarithmic units for subsequent analysis.

### 2.4. Density Modelling

The International System of Units established seven main units: meter, kilogram, second, kelvin, ampere, candela, and mole, corresponding to the quantities of length, mass, time, temperature, the intensity of electric current, luminous intensity, and amount of substance, respectively. From these seven, several secondary units and magnitudes are derived, including density, represented by the Greek letter *ρ*. Density describes the amount of mass of any material that occupies a space; hence, the density of a material element can be obtained in a volumetric, surface, and linear system, whose units are described as *ρ* ⇐ kg/m^3^, *ρ*_S_ ⇐ kg/m^2^, and *ρ*_L_ ⇐ kg/m. Density is represented with the following expression:$$\rho =\frac{m}{v}$$
where *ρ* is the volumetric density, *m* is the mass of the material, and *v* is the volume. From this expression, you can adjust the multiples and submultiples of the units and their corresponding unit conversions. In this context, the expression of density is applied to determine the conversion of population density units to microbiological surface and volumetric densities., as described in Table 1.

### 2.5. Statistical Analysis

The experiment design was a completely randomized one-way design with five replicates per treatment and the blocking criterion was the meat batch. Data were analyzed using the ANOVA and MANOVA procedures with the General Lineal Model (GLM) procedure in SAS^®^ software, version 9.1.3 (SAS Institute Inc., Cary, NC, USA, 2006). Subsequently, a multiple comparison of means was carried out by the Tukey test. The statistical model is represented by the equation:y_ijk_ = μ + τ_i_ + *β_j_* + Ɛ_ijk_
where y_ijk_ is the responding variable measured in the kj-th repetition of the i-th treatment and the j-th block, τ_i_ is the effect of the i-th treatment, *β_j_* is the effect of the j-th batch, and Ɛ_ijk_ is the random error corresponding to the kj-th repetition of the i-th treatment and the j-th block. The Ɛ_ijk_ are assumed to be identically and independently normally distributed, with a mean of zero and a variance of σ^2^_ε_.

## 3. Results and Discussion

Fresh or recently obtained bovine and ovine carcasses typically have mesophilic organisms between 2 and 4 Log_10_CFU/cm^2^; a quantity greater than 5 Log_10_CFU/cm^2^ indicates deficiencies in the sanitary handling of the abattoir and the carcass [9,10]. In general, the microbial numbers on the meat surface can be classified as high when it is in a range of 5–7 Log_10_CFU/cm^2^, intermediate when it is between 3 and 4 Log_10_CFU/cm^2^, and low when it is below 3 Log_10_CFU/cm^2^ [11]. According to the above, it can be inferred that the meat used in the present study resulted from good sanitary management since its microbial numbers were classified between intermediate and low, as it was below 4 Log_10_CFU/cm^2^, except for *B. thermosphacta* with the DP protocol, 5.12 ± 0.76 Log_10_CFU/mL, which is considered high (Table 1 and Table 2).

The number of LAB after 72 h of maturation did not exceed 4 Log_10_CFU/cm^2^ in any of the sampling protocols, with values of 2.29 ± 0.59, 2.57 ± 0.86, and 3.91 ± 0.74 Log_10_CFU for SW, EX, and DP, respectively; this is probably because the maturation time was shorter, compared to the shelf life in other countries [12]. Vacuum packaging is a method of distribution and marketing where meat is kept at −1 °C in oxygen-tight bags to increase the shelf life [12]. If the meat is vacuum-packed under hygienic processing and packaging conditions, the number of LAB is normally low (<2 Log_10_CFU/cm^2^); however, this number tends to increase during storage, exceeding 6 Log_10_CFU/cm^2^ after 2 or 3 weeks of storage [13,14].

The numbers of viable microorganisms had significant differences (*p* < 0.05) between the sampling methods. The microorganisms recovered in DP were significantly higher (*p* < 0.05) than EX and SW. The SW and EX methods had no statistical differences between them (*p* > 0.05), except for *B. thermosphacta* (3.73 ± 0.75 and 3.01 ± 0.88 Log_10_CFU/cm^2^) and *Enterobacteriaceae* (2.63 ± 0.98 and 1.95 ± 0.55 Log_10_CFU/cm^2^), where the number of microorganisms was higher in EX.

Results have shown that SW was the least effective method for the recovery of microorganisms, probably because only part of the microflora is recovered [15], since it has been reported that there is a bidirectional transfer of microorganisms from the product to the swab and from the swab to the product. Therefore, it could be assumed that a percentage of the microorganisms remains on the surface after sampling [16]. On the other hand, the fat from the meat can clog the pores of the swab as the sample is collected, which could result in the partial recovery of microorganisms [17]. Although swabbing is considered a basic method and is one of the most widely used methods for assessing the level of surface microorganisms, its efficiency is greater on flat and smooth surfaces [18]. It is important to mention that to increase the number of microorganisms recovered by the swabbing method, it is recommended that the samples cover an approximate area of 600 cm^2^ [19]. Although swabbing has a low recovery rate, it is preferred over methods such as rinsing or excision, as it is a non-invasive and non-destructive method and is practical to perform throughout the production process [15,16,20].

The excision method is the most recommended by standards because it is usually the most effective for the recovery of microorganisms, due to the total extraction of the surface of the meat; however, it is the most exposed to contamination [7,21]. Excision is also considered to be the least variable type of sampling due to the complete recovery of microorganisms that are firmly attached to the surface [7,20,22,23]. The number of microorganisms recovered by excision is usually higher than swabbing [24,25].

Excision and rinsing typically have the same percentage of recovery of microorganisms, and both methods are superior to swabbing [15,26]. Contrarily, it has been observed that the recovery of *Enterobacteriaceae* was higher by the rinsing method, followed by excision and finally swabbing [8,27].

When calculating the recovery ratio between sampling methods, it became evident that for each colony-forming unit recovered by excision and swabbing, between 1.3 and 1.9 Log_10_ CFU more could be recovered by the drip method for EX vs. DP (*Salmonella*) and SW vs. DP (*Enterobacteriaceae*), respectively (Table 3). However, a relationship greater than 2 Log_10_CFU has been reported in vacuum-packed meat stored for longer periods [28]. So, if the intention is to microbiologically analyze vacuum-packed meat, the DP sampling option should be considered.

## 4. Conclusions

This research compared the number of microorganisms recovered from vacuum-packed beef using three sampling methods (swabbing, excision, and drip). Swabbing and excision are the most-used sampling methods in the meat industry and recommended by national and international regulations. Despite not being considered as a sampling method in various regulations, the drip method was the one that recovered the largest number of microorganisms. However, the drip method is limited to packaged meat containing exudate despite its high degree of microbial recovery. It is important to consider the above when designing and implementing a sampling system for the meat industry, so that it reflects the microbial load of the meat, considering that for each bacterium enumerated by the excision or smear methods, there could be up to two logarithms more microorganisms.

## Figures and Tables

**Table 1 microorganisms-13-00159-t001:** Numbers of microorganisms (mean ± standard deviation) including unit conversions applied to surface and volumetric density.

Microorganism	Sampling Method ^1^		
	Swabbing (gr/mm^2^)	Excision (gr/mm^2^)	Drip (gr/mm^3^)
*Brochothrix thermosphacta*	0.397 × 10^−18 c^	2.083 × 10^−18 b^	5.114 × 10^−18 a^
*Salmonella* spp.	64.913 × 10^−21 b^	1002.88 × 10^−21 b^	31.069 × 10^−21 a^
*Enterobacteriaceae*	0.891 × 10^−12 c^	4.265 × 10^−12 b^	4.786 × 10^−12 a^
Lactic acid bacteria	235 × 10^−6 b^	449 × 10^−6 b^	9.827 × 10^−3 a^
Yeasts and moulds	1.138 × 10^−3 b^	4.744 × 10^−3 b^	4.854 × 10^−6 a^

^1^ Means in the same line with different letters are statistically different (*p* < 0.05).

**Table 2 microorganisms-13-00159-t002:** Numbers of microorganisms (mean ± standard deviation) recovered from vacuum-packed beef following three sampling protocols.

Microorganism ^1^	Sampling Method ^1^		
	Swabbing (Log_10_CFU/cm^2^)	Excision (Log_10_CFU/cm^2^)	Drip (Log_10_CFU/mL)
*Brochothrix thermosphacta*	3.01 ± 0.88 ^c^	3.73 ± 0.75 ^b^	5.12± 0.76 ^a^
*Salmonella* spp.	1.70 ± 0.00 ^b^	1.90 ± 0.53 ^b^	2.38 ± 0.81 ^a^
*Enterobacteriaceae*	1.95± 0.55 ^c^	2.63 ± 0.98 ^b^	3.68 ± 1.02 ^a^
Lactic acid bacteria	2.29 ± 0.59 ^b^	2.57 ± 0.86 ^b^	3.91 ± 0.74 ^a^
Yeasts and moulds	2.25 ± 1.53 ^b^	2.87 ± 0.87 ^b^	3.88 ± 1.44 ^a^

^1^ CFU = colony-forming units. Means in the same line with different letters are statistically different (*p* < 0.05).

**Table 3 microorganisms-13-00159-t003:** Relationship of the number of microorganisms recovered by three sampling methods of vacuum-packed meat.

Microorganism	Ratio of Sampling Methods by Number of Microorganisms Recovered (Log_10_ CFU)		
	Excision vs. Drip	Swabbing vs. Drip	Swabbing vs. Excision
*Brochothrix thermosphacta*	1:1.4	1:1.7	1:1.2
*Salmonella* spp.	1:1.3	1:1.4	1:1.1
*Enterobacteriaceae*	1:1.4	1:1.9	1:1.3
Lactic acid bacteria	1:1.5	1:1.7	1:1.1
Yeasts and moulds	1:1.4	1:1.7	1:1.1

CFU = colony-forming units.

## Data Availability

Data are contained within the article.

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
