# Peer review of "The Swab, the Drip, or the Meat? Comparison of Microbiological Sampling Methods in Vacuum-Packed Raw Beef"

_microorganisms, 2025, doi:10.3390/microorganisms13010159_

Round 1
Reviewer 1 Report
Comments and Suggestions for Authors
The present research titled "The Swab, the drip, or the meat? Comparison of microbiological sampling methods in vacuum-packed raw beef" is a very interesting study with significant practical applications, aligning well with the journal's scope. However, there are several major issues that need to be addressed to enhance the rigor and clarity of the manuscript.
Major issues
1. While the authors state that the experiment was laid out in a completely randomized design, it remains unclear how the samples were assigned to treatments. How were variations in prior contaminations accounted for before assigning samples to each treatment? It would be important to include controls for contamination during sample preparation and sampling for each treatment.
2. Please provide details regarding the dilutions used for colony counting to ensure reproducibility.
3. I recommend incorporating qPCR to validate and ensure the reliability of the results.
Comments on the Quality of English Language1. There are long sentences in the discussion section that may be difficult to understand.
2. grammar issues: "it is laid out?
Author Response
Reviewer comment 1. While the authors state that the experiment was laid out in a completely randomized design, it remains unclear how the samples were assigned to treatments.
Answer. Every meat pack (samples) was sampled with the three protocols in the following order; first drip, then swabbing and finally excision. The comment is highly appreciated as it was not stated before in the paper, but now it is included in the description.
Reviewer comment 2. It would be important to include controls for contamination during sample preparation and sampling for each treatment.
Answer. Beef, naturally has between 2 - 4 log10 CFU, hence, the meat packs were prepared and sampled under controlled conditions (in an inoculation chamber).
Reviewer comment 3. Please provide details regarding the dilutions used for colony counting to ensure reproducibility.
Answer. This section was modified to explain the dilution and inoculation methods.
Reviewer comment 4. I recommend incorporating qPCR to validate and ensure the reliability of the results.
Answer. The comment is highly appreciated and agreed that such method would add a higher level of reliability. Unfortunately, as we do not have the resources to perform QPCR, the issue was approached only through traditional microbiology methods.
Reviewer comment 5. There are long sentences in the discussion section that may be difficult to understand.
Answer. The whole section was checked.
Reviewer 2 Report
Comments and Suggestions for Authors
Dear Authors,
The work is written precisely, without superfluous details. The structure is appropriate. The literature is not the most up-to-date, there are almost no references from the last 5 years. Regardless of the fact that it is an original Article, the number of literature references is scarce. The proposal is to expand the list of references with more recent references. The literature in the list of references is cited in a way that is NOT in accordance with the Instructions for Authors, so the whole must be redacted. I entered all the above suggestions and needs/reasons for correction as comments in the PDF document that I downloaded from the SuSy system, so that they are visible to the authors, other reviewers and editors.

Author Response
Comment 1. Perhaps it would be more correct to write "microbiological status", instead of "microbiological conditions".
It has been modified as suggested.
Comment 2. .... "which is essential for the unification of measurement standards and conversion factors".
Maybe a more precise word is "unification" that "homogenizing"
It has been modified as suggested.
Comment 3. Is it Musculus semimembranosus? Is it Beef International SA de CV a retail chain or something else?
It has been modified as suggested.
Comment 4. 70% ethyl alcohol
It has been modified as suggested.
Comment 5. mL - the symbol for a liter is written with a capital letter (L) according to the valid nomenclature.
It has been modified as suggested.
Comment 6. small letter: w
It has been modified as suggested.
Comment 7. mL
Modified as suggested.
Comment 8. letters? Literals means and typo too
Modified as suggested.
Comment 9. Throughout the whole table mark as a multiplication symbol (×).
Modified as suggested.
Comment 10. Insert correct symbol for degrees.
Modified as suggested.
Comment 11. 15,16,20
Modified as suggested.
Comment 12. I ask the authors not to leave a space between the comma and the next digit when citing reference numbers, in accordance with the instructions for authors.
In the entire text of the manuscript, it is understood.
Modified as suggested.
Comment 13. 8, 27
Modified as suggested.
Comment 14. All citations (references) must be redacted according to the instructions for authors (authors are separated with ; the name of the journal must be in italics and as an abbreviation, the year of publication in bold, the volume in italics).
Modified as suggested.
Comment 15. Names of microorganisms (genus, species - binomial name) must be in italic font, in all places in the text of the manuscript as well as in references.
Modified as suggested.
Comment 16. The literature is not the most up-to-date, there are almost no references from the last 5 years. Regardless of the fact that it is an original Article, the number of literature references is scarce. The proposal is to expand the list of references with more recent references.
The review and comments are fully appreciated. All the comments were modified as suggested, except for the inclusion of recent references. As no references were found that could be related to the topic presented.
Round 2
Reviewer 1 Report
Comments and Suggestions for Authors
The authors have revised the manuscript per my suggestions